# Silicosis and Pulmonary Functions Among Residents Exposed to Dust in Saraburi Thailand

**DOI:** 10.3390/diseases13110372

**Published:** 2025-11-13

**Authors:** Narongkorn Saiphoklang, Pitchayapa Ruchiwit, Apichart Kanitsap, Pichaya Tantiyavarong, Pasitpon Vatcharavongvan, Srimuang Palungrit, Kanyada Leelasittikul, Apiwat Pugongchai, Orapan Poachanukoon

**Affiliations:** 1Department of Internal Medicine, Faculty of Medicine, Thammasat University, Pathum Thani 12120, Thailand; 2Medical Diagnostics Unit, Thammasat University Hospital, Pathum Thani 12120, Thailand; 3Center of Excellence for Allergy, Asthma and Pulmonary Diseases, Thammasat University Hospital, Pathum Thani 12120, Thailand; 4Department of Clinical Epidemiology, Faculty of Medicine, Thammasat University, Pathum Thani 12120, Thailand; 5Department of Community Medicine and Family Medicine, Faculty of Medicine, Thammasat University, Pathum Thani 12120, Thailand; 6Department of Pediatrics, Faculty of Medicine, Thammasat University, Pathum Thani 12120, Thailand

**Keywords:** age, asthma, chronic obstructive pulmonary disease, pulmonary function, restrictive defect, silicosis

## Abstract

**Background**: Silicosis is a lung disease caused by inhalation of crystalline silica dust, leading to lung fibrosis, respiratory symptoms, and impaired lung function. This study aimed to determine the prevalence of silicosis, asthma, and chronic obstructive pulmonary disease (COPD), and to identify factors associated with abnormal pulmonary function among residents living in dust-exposed areas in Thailand. **Methods:** A cross-sectional study was conducted from March 2024 to July 2024 among adults aged 18 years or older in Saraburi, Thailand. Data collected included demographics, comorbidities, respiratory symptoms, risk of silicosis, chest radiographs, and spirometry (forced vital capacity (FVC), forced expiratory volume in one second (FEV_1_), and bronchodilator responsiveness (BDR)). Silicosis was confirmed based on a history of significant silica exposure and characteristic chest radiographic findings. **Results:** Among 290 participants (55.9% female, mean age 47.6 ± 16.4 years), the prevalence of silicosis, asthma, and COPD was 0.3%, 4.5%, and 10.3%, respectively. Abnormal chest radiographs were observed in 8.3%, and abnormal lung function in 34.1%, including restrictive lung patterns (16.6%), airway obstruction (9.0%), mixed defects (2.8%), and small-airway disease (5.9%). BDR was observed in 4.8%. Logistic regression identified increasing age as a significant predictor of abnormal lung function. **Conclusions:** Silicosis prevalence was lower than that of asthma and COPD, but abnormal pulmonary function—especially restrictive defects—was common. Notably, the prevalence of asthma and COPD was higher than previously reported community-based diagnosis rates, suggesting potential underdiagnosis. Older age was associated with a higher likelihood abnormal lung function. These findings highlight the need for targeted surveillance, preventive measures, and public health interventions to mitigate the respiratory impacts of dust exposure in community settings

## 1. Introduction

Silicosis is a preventable occupational lung disease caused by inhaling crystalline silica and other fine sand dust particles [1,2]. This exposure leads to small fibrotic nodules in the lungs, resulting in restrictive ventilatory defects and impaired gas transfer [3]. The severity of restriction and gas-transfer impairment correlates with radiographic extent of disease, particularly in cases of progressive massive fibrosis, and is associated with poorer outcomes, including higher risk of mortality [4]. Common symptoms of silicosis include shortness of breath, cough, and fatigue without fever [5]. In advanced cases, chest X-rays reveal large, symmetric, bilateral opacities with irregular margins, often in the middle lung zones or the peripheral one-third of the lungs [6]. The presence of radiographic hyperinflation is strongly associated with progressive functional deterioration [7]. Overall, silicosis contributes significantly to lung function decline [8,9].

Airway obstruction is also a consequence of silica dust exposure [2]. Cumulative exposure can cause airflow obstruction and reduced pulmonary function, even in the absence of evidence of radiographic silicosis [10,11]. Smoking, dust exposure, and emphysema are key confounding factors when evaluating the relationship between silicosis and lung function [12]. Occupational silica exposure, with or without radiographic silicosis, is a recognized non-tobacco cause of chronic obstructive pulmonary disease (COPD), which includes chronic bronchitis, emphysema and small-airways disease [13]. The risk of COPD is approximately 1.5 times higher in dust-exposed workers compared with controls [14]. In addition, silica exposure—and in some cases silicosis—has been linked to asthma or asthma-like occupational airway disease, although the relationship remains indeterminate [15,16,17].

Occupations at risk for developing silicosis include stone carvers, miners, and workers engaged in cutting, quarrying, crushing, grinding, or tunnelling rock, sand, bricks, tiles, or concrete—activities that generate respirable crystalline silica (RCS) dust [18,19]. Other high-risk industries include glass, ceramics, refractory materials, foundry work, pottery, and sanitary ware manufacturing, where processing silica-containing materials releases hazardous dust [19,20]. Dust exposure is consistently associated with reduced lung function, manifesting as both obstructive and restrictive patterns depending on dust type and exposure profile [21]. Several international community- and population-level studies—including those conducted in industrial or ceramic regions and areas with high ambient particulate matter—have shown that environmental dust and particulate pollution are linked to poorer lung function in non-occupational populations, including both children and adults [22]. Systematic reviews and meta-analyses across various dust types (organic dusts, inorganic particulate matter, and silica) further support the overall negative impact of dust exposure on pulmonary function, although the magnitude of the effect varies according to dust composition, exposure intensity, use of protective measures, and study design [23].

Data on silicosis and airway diseases from community-based studies of dust-exposed regions of Thailand remain limited, particularly in Saraburi Province, where most previous research has focused on specific occupational groups such as stone carvers or ceramic workers. It was hypothesized that the prevalence of silicosis in this community-based population would be similar to that previously reported among Thai stone carvers (36.1%) [24], reflecting the potential respiratory impact of environmental dust exposure in the region. Therefore, this study aimed to determine the prevalence of silicosis, asthma, and COPD and to identify factors associated with abnormal pulmonary function among local residents.

## 2. Materials and Methods

### 2.1. Study Design and Participants

A cross-sectional study was conducted among residents of Phra Phutthabat District, Saraburi Province, Thailand, between March 2024 and July 2024. Phra Phutthabat, located in central Thailand, is approximately 140 km from Bangkok. Long-established mining and cement industries are widespread in this area. Individuals aged 18 years or older were included. Exclusion criteria were recent myocardial infarction within the past three months, blood pressure higher than 180/100 mmHg, resting heart rate greater than 120 beats per minute, pregnancy, and inability to perform spirometry.

Participants were recruited using a convenience sampling approach. Residents living in Phra Phutthabat District were invited to participate through community announcements and local health centers. All eligible individuals who provided informed consent were enrolled consecutively during the study period.

The study protocol was approved by the Human Research Ethics Committee of Thammasat University (Medicine) (IRB No. MTU-EC-IM-0-300/65, COA No. 145/2023 Date of approval: 13 June 2023). All participants provided written informed consent. This study was prospectively registered with Thaiclinicaltrials.org with number TCTR20230713002.

### 2.2. Procedures and Outcomes

Demographic data, pre-existing comorbidities, respiratory symptoms, risk of silicosis, and lung function measurements by spirometry—including forced vital capacity (FVC), forced expiratory volume in one second (FEV_1_), peak expiration flow (PEF), forced expiration flow rate at 25–75% of FVC (FEF_25–75_), and bronchodilator responsiveness (BDR)—were collected. Spirometry was performed according to the international guidelines of the American Thoracic Society and European Respiratory Society [25,26,27] using PC-based spirometer (Vyntus SPIRO, Vyaire Medical, Mettawa, IL, USA). The spirometer was calibrated daily with a 3-L syringe before testing each day to ensure accuracy. All tests were performed by trained technicians who had received standardized instruction in spirometry procedures. Regular supervision and review of test quality were performed by a senior pulmonologist to minimize inter-operator variability. Participants were instructed to exhale into the mouthpiece as forcefully and as long as possible (at least 15 s). FVC, FEV_1_, FEV_1_/FVC, PEF and FEF_25–75_ were reported in liters (L), percent predicted (%pred), %, or liters per second (L/s). BDR was tested by administering 400 µg of salbutamol, followed by repeat spirometry after 15 min [25,26,27]. Predicted values of all spirometry parameters were calculated according to the Global Lung Function Initiative reference equations [28].

Dust exposure was assessed through a structured questionnaire addressing both occupational and environmental sources. For occupational exposure, participants were asked about current or past employment in industries with known dust exposure (e.g., stone cutting, quarrying, cement, or ceramic production), as well as the duration of employment (in years). For non-occupational environmental exposure, participants were asked about residential proximity to dust-generating sites such as quarries or cement plants, as well as the duration of residence in the area. Exposure was classified as occupational, environmental, or none based on these responses. Quantitative measurements of dust concentration were not available in this community-based study.

Abnormal lung function was assessed using standardized criteria: airway obstruction was defined as FEV_1_/FVC ratio below lower limit of normal (LLN); restrictive defect was defined as FEV_1_/FVC above LLN with FVC below LLN [29]; mixed obstructive and restrictive defect was defined as FEV_1_/FVC ratio below LLN with FVC below LLN; and small-airway disease was defined as FEF_25–75_ below 65% [30]. BDR was defined as an improvement in FEV_1_ or FVC greater than 10% of the predicted value after bronchodilator administration [29]. Abnormal lung function was recorded at baseline.

In this study, airway diseases were classified as COPD or asthma. COPD was defined as having respiratory symptoms (cough, sputum production, dyspnea, or wheezing), relevant risk factors (particularly smoking ≥10 pack-years or biomass fuel use), and a post-bronchodilator FEV_1_/FVC < 70% [31]. Asthma was defined as having respiratory symptoms (wheezing, dyspnea, chest tightness, or cough) with a positive BDR [32]. Silicosis was defined as documented exposure to RCS dust (e.g., mining, stone cutting, foundry work, construction, sandblasting) for ≥1 year, chest radiographs consistent with an International Labour Organization (ILO) profusion category of ≥1/0, and exclusion of other possible causes of pulmonary nodules (e.g., pulmonary tuberculosis) [3].

Chest radiographs were obtained in the standard posteroanterior (PA) upright position and classified according to ILO system for pneumoconiosis [3]. All films were independently reviewed by two certified radiologists who were blinded to participants’ clinical data. Any discrepancies were resolved by consensus. Data entry quality control was maintained through double-entry verification and periodic cross-checks with source documents to minimize transcription errors.

### 2.3. Statistical Analysis

The sample size was calculated based on a previous study reporting a 36.1% prevalence of silicosis among stone carving workers in Nakhon Ratchasima Province, Thailand [24]. A similar prevalence was assumed in this population. The calculation was performed to estimate this proportion with 80% statistical power, a Type I error (α) of 5%, and a precision margin of 5%, yielding a required sample size of 355.

Categorical variables were expressed as numbers (percentages), and continuous variables as mean ± standard deviation. The chi-squared test was used to compare categorical data between the abnormal and normal lung function groups, and Student’s *t*-test was used to compare the means of continuous data between the two groups. Logistic regression was applied to identify factors associated with abnormal lung function, with age, smoking, comorbidities, and respiratory symptoms included if they were significant in bivariate analysis (*p*-value < 0.05) or had a significantly impact on pulmonary function. Adjusted odds ratios (ORs) with 95% confidence intervals (CI) were reported. A two-sided *p*-value < 0.05 was considered statistically significant. Analyses were performed using SPSS version 26.0 software (IBM Corp., Armonk, NY, USA).

## 3. Results

### 3.1. Participants

A total of 302 participants were screened. Of these, 290 were included in the study (55.9% female), while 12 were excluded (Figure 1). The mean age was 47.6 ± 16.4 years. Current or former smokers accounted for 29.7%, with an average smoking history of 2.7 ± 8.6 pack-years. Hypertension (25.9%) and hyperlipidemia (17.6%) were common pre-existing comorbidities. The most common respiratory symptoms were cough (60.4%), breathlessness (13.8%), and sputum production 20.3%) (Table 1).

### 3.2. Outcomes

The mean FEV_1_ was 90.4 ± 15.9% of the predicted value, FVC was 92.2 ± 14.9% of the predicted value, and the mean FEV_1_/FVC was 81.99 ± 8.40% (Table 2, Figure 2 and Figure 3). Abnormal pulmonary functions were observed in 34.1% of participants (Table 3), including restrictive defect (16.6%), airway obstruction (9.0%), mixed defect (2.8%), and small-airway disease (5.9%). BDR was present in 4.8% (Table 3). Abnormal pulmonary function patterns across age groups are demonstrated in Figure 4. The prevalence of asthma and COPD was 4.5% and 10.3%, respectively (Table 2).

Abnormal chest radiographic findings were observed in 8.3% of participants, including lung fibrosis (2.8%), pleural fibrosis (2.1%), lung nodules (2.1%), mixed patterns (2.8%), and silicotic patterns (0.3%) (Table 3). The prevalence of silicosis was 0.3%.

Regarding the identified silicosis case, the participant was a 37-year-old woman with a 10-year history of cement factory exposure who had lived in the area for 20 years, with no history of pulmonary tuberculosis. Her spirometry showed an FEV_1_ of 1.21 L (43% predicted), an FVC of 1.60 L (48% predicted), and an FEV_1_/FVC ratio of 0.76, indicating a restrictive pattern. Chest radiography demonstrated a moderate amount of small rounded opacities predominantly in the upper lung zones, consistent with ILO category 2/2.

Participants with abnormal pulmonary function were significantly older and had higher rates of coronary heart disease, cough, and breathlessness compared to those with normal function (Table 1). Logistic regression analysis indicated that increasing age was associated with a higher risk of abnormal lung function (Table 4). Among abnormal pulmonary function patterns, only the obstructive pattern was associated with increasing age compared to the non-obstructive group (58.6 ± 11.8 years vs. 46.5 ± 16.4 years, *p* < 0.001).

## 4. Discussion

This is the first study to assess the prevalence of silicosis, asthma, and COPD, as well as factors associated with abnormal pulmonary function, among residents in dust-exposed areas in Saraburi Province, Thailand, rather than in specific occupational groups such as stone carvers or ceramic workers. Silicosis was found in 0.3% of participants, asthma in 4.5%, and COPD in 10.3%, with 34.1% exhibiting abnormal lung function—mostly restrictive defects (16.6%). The participant with silicosis is a 37-year-old female who had lived in the area for 20 years and worked in a cement factory for 10 years, with no history of pulmonary tuberculosis. Although only one case of silicosis (0.3%) was identified, this finding is notable because it occurred in a relatively young participant (37 years old) with long-term residential and occupational exposure to silica dust. This suggests that, despite the low prevalence, ongoing dust exposure remains a potential risk for silicosis development in this community. The low rate may reflect improved occupational safety or under-recognition of early disease stages, as chest radiography has limited sensitivity for early silicosis.

Silicosis is a well-recognized occupational lung disease caused by inhalation of crystalline silica dust, resulting in irreversible lung scarring and nodular lesions. Although incurable, it is preventable through effective dust control and occupational safeguards [1,2]. Previous studies in Thailand reported higher prevalence: 36.1% among stone-carving workers in Sikhiu District, Nakhon Ratchasima Province [24], and 2.9% among ceramic workers after chest radiograph reinterpretation [33]. In contrast, this study found a much lower prevalence of 0.3% among community residents, likely reflecting lower occupational dust exposure. Nevertheless, this underscores the persistent presence of silicosis in Thailand and highlights the need for prevention and early surveillance, even in non-occupational settings. The lower rates among residents compared with formal workers likely reflects differences in exposure intensity, but the detection of silicosis—even if rare—emphasizes the importance of strengthening occupational and environmental monitoring of silica exposure, particularly in informal or community settings.

In the study of spirometry data in subjects with silicosis conducted by Yang S et al. in Hong Kong [4], it was found that the prevalence of a restrictive pattern was 24.1%, which included 11.0% with a restrictive pattern only and 13.1% with a mixed restrictive pattern and airflow obstruction. Additionally, a case of silicosis in our study also showed a restrictive pattern. Finally, reliance on chest radiography likely underestimates the true prevalence of early or mild silicosis. Recent studies and systematic reviews show that high-resolution computed tomography (HRCT) detects substantially more cases and finer parenchymal changes than standard chest X-rays, with higher sensitivity for early silicosis and accelerated silicosis in artificial stone workers [34,35,36]. Therefore, our use of chest radiography should be interpreted as a conservative estimate of radiographic silicosis burden.

Beyond silicosis, the relatively high prevalence of COPD (10.3%) and asthma (4.5%) highlights the broader respiratory consequences of dust exposure. Chronic inhalation of silica and other particulates may induce airway inflammation and remodeling, contributing to both obstructive and restrictive patterns. This aligns with the 34.1% of participants who demonstrated abnormal lung function, predominantly restrictive defects. These findings suggest that even in the absence of radiographic silicosis, dust exposure may lead to functional respiratory impairment warranting ongoing surveillance and preventive measures.

Asthma prevalence varies widely worldwide, with occupational dust exposure contributing significantly to its burden [37,38]. Although community-level data in Thailand are limited, the observed prevalence of 4.5% aligns with global ranges for general populations. This is lower than the 10.3% reported in a recent national asthma and COPD survey by Saiphoklang N et al. [39]. Given the role of occupational and environmental dust in asthma development [37], these findings likely reflect, at least in part, environmental dust exposure in the study communities. Environmental dust can contribute to the development of asthma, particularly in children, as it may contain allergens such as dust mites, fungal spores, and other particles that trigger inflammation and allergic reactions in the airways [40,41].

COPD was observed in 10.3% of participants, a rate higher than previous national and regional estimates. The 1999 national survey remains the most comprehensive population-based study in Thailand, reporting a prevalence of 2.1% with projected increases to 7.0% by 2010 [42]. Although no newer nationwide data are available, a recent survey by Saiphoklang N et al. reported a prevalence of 8.3% in 2025 [39], and a regional study in Chiang Mai found a prevalence of 5–7% in 2015, with rural areas at 6.8% and an underdiagnosis rate of 80–85% [43]. The higher prevalence may reflect under-recognition in prior surveys or increased risk from dust exposure in rural and occupational settings. Environmental and occupational dusts, such as coal dust, grain dust, and silica dust, can increase the risk of developing COPD [14]. The risk is even higher if combined with other factors like smoking, and both the concentration and duration of dust exposure play significant roles in disease development [14]. A previous study of stone-grinding factories in Saraburi, Thailand, found average levels of total dust and respirable crystalline silica (RCS) dust of 24.3 ± 34.6 mg/m^3^ and 2.4 ± 1.6 mg/m^3^, respectively, with silicosis observed in 9% of workers [44]. However, reports of environmental dust in this area remain limited.

Furthermore, the prevalence of asthma and COPD in this study (4.5% and 10.3%, respectively) exceeded the rates of pre-existing asthma and COPD (3.1% and 0%, respectively), indicating new diagnoses among residents in dust-exposed areas. This finding highlights the underdiagnosis of these airway diseases and underscores the need for screening with spirometry among residents living in such environments.

Altered pulmonary function is common among dust-exposed workers. A Chinese study of silica-exposed workers reported obstructive dysfunction (FEV_1_/FVC < 70%) in 2.3% and restrictive dysfunction in 8.1% [45], consistent with findings that restrictive defects were the most prevalent pulmonary abnormality (16.6%). This suggests that dust exposure, along with aging and comorbidities, may contribute more to restrictive than obstructive impairment. The predominance of restrictive defects in this population reflects patterns seen in silica-exposed cohorts and highlights the broader impact of dust on lung function beyond classic obstructive disease.

Consistent with these findings, age is a recognized risk factor for lung function decline and COPD [42,45], and dust exposure may exacerbate age-related risks, particularly for restrictive pathology. Prevention may be a key strategy to reduce the development of these respiratory abnormalities in dust-exposed communities.

This study has several limitations. First, the sample size is very small with only one case of silicosis observed. Second, there is a lack of data on environmental dust concentration data in this study area, which limits the quantitative assessment of exposure and its relationship with outcomes. Third, although key demographic and smoking-related factors were adjusted for, the possibility of unmeasured or residual confounding remains. In particular, body mass index and exposure to biomass fuel smoke were not comprehensively captured in our study. Both factors have been associated with impaired lung function and an increased risk of chronic respiratory disease in community settings. The absence of detailed data on these exposures may therefore have led to an underestimation or overestimation of their contribution to the observed lung function abnormalities. Fourth, advanced lung function assessments, such as lung volume measurement, diffusing capacity of the lungs for carbon monoxide (DLCO), and impulse oscillometry (IOS), were not performed, which may have led to an underestimation of pulmonary abnormalities. Fifth, chest radiography was the sole imaging modality used for radiologic assessment. Chest X-ray has lower sensitivity than HRCT for detecting early and mild silicosis; consequently, the true prevalence of silicosis in this community may be higher than reported. Finally, the absence of long-term follow-up limits evaluation of disease progression. Prospective studies are needed to assess changes in lung function, radiographic findings, and clinical outcomes over time in this population.

## 5. Conclusions

In this community-based population, although silicosis was rare, the presence of obstructive and restrictive lung function abnormalities in over one-third of participants underscores the broader respiratory impact of environmental dust exposure. The prevalence of COPD and asthma exceeded national estimates, suggesting that chronic dust exposure may contribute to airway inflammation and functional impairment even in the absence of radiographic silicosis. These findings highlight the burden of non-communicable respiratory diseases in dust-exposed areas and emphasize the need for ongoing surveillance, preventive measures, and early interventions to mitigate long-term respiratory health impacts.

However, the study was limited by its cross-sectional design, reliance on chest radiography for silicosis detection, and potential selection bias from voluntary participation. Longitudinal studies using high-resolution imaging and quantitative exposure assessment are warranted to clarify causal relationships and identify early subclinical changes.

Future studies should explore biomarkers of early silica-induced lung injury and evaluate the long-term respiratory outcomes of environmental dust exposure. Community-based dust control interventions and regular pulmonary function screening should be implemented to reduce respiratory health risks in exposed populations.

## Figures and Tables

**Figure 1 diseases-13-00372-f001:**
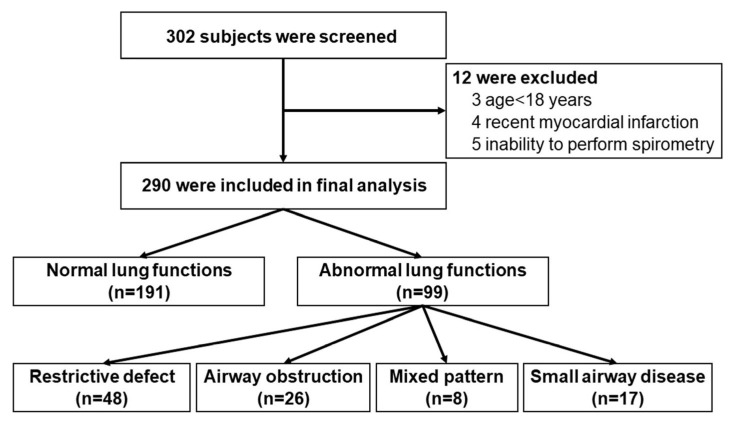
Flowchart of participant recruitment to the study.

**Figure 2 diseases-13-00372-f002:**
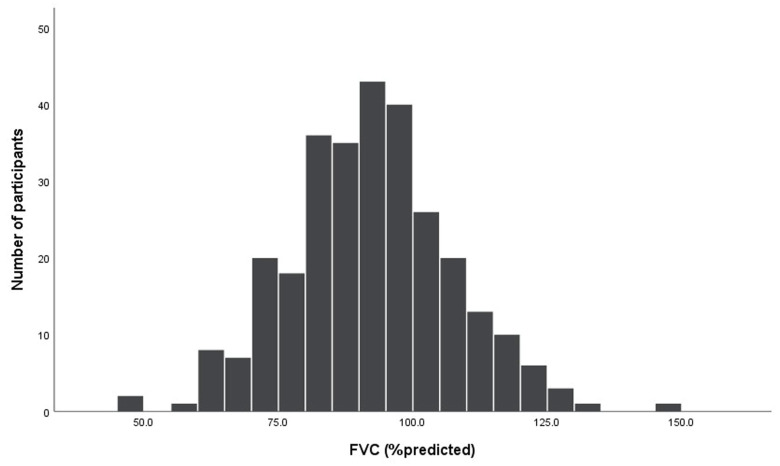
Histogram of % predicted values of forced vital capacity (FVC) for all participants (n = 290).

**Figure 3 diseases-13-00372-f003:**
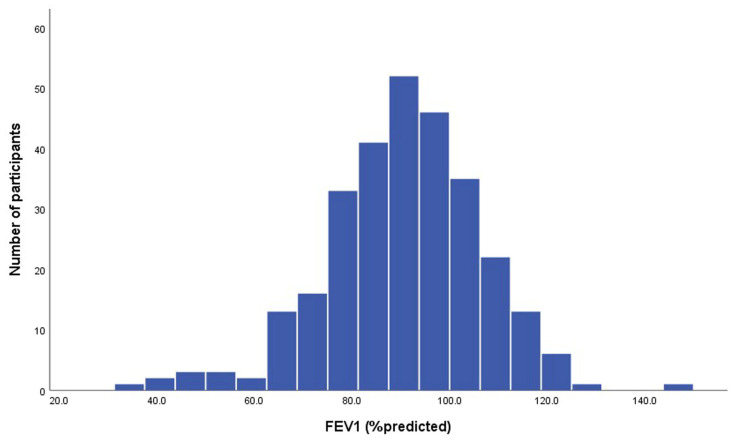
Histogram of % predicted values of forced expiratory volume in one second (FEV_1_) for all participants (n = 290).

**Figure 4 diseases-13-00372-f004:**
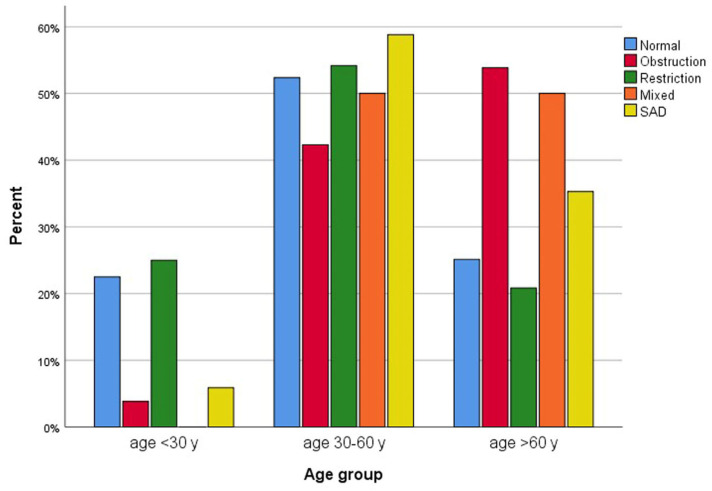
Bar chart of pulmonary function across age groups (n = 290). SAD = small-airway disease; y = years; mixed = mixed obstructive and restrictive defect.

**Table 1 diseases-13-00372-t001:** Baseline characteristics of participants.

Characteristics	Total (n = 290)	Normal Lung Function (n = 191)	Abnormal Lung Function (n = 99)	*p*-Value
Age, years	47.6 ± 16.4	45.9 ± 16.6	50.9 ± 15.6	0.013
Female	162 (55.9)	111 (58.1)	51 (51.5)	0.283
Male	128 (44.1)	80 (41.9)	48 (48.5)	0.283
Body mass index, kg/m^2^	25.2 ± 5.2	25.2 ± 4.7	25.2 ± 6.2	0.957
Smoking	86 (29.7)	34 (34.3)	52 (27.2)	0.208
Amount of smoking, pack-years	9.7 ± 14.0	8.4 ± 13.6	11.5 ± 14.5	0.329
**Pre-existing comorbidities**				
Hypertension	75 (25.9)	47 (24.6)	28 (28.3)	0.498
Hyperlipidemia	51 (17.6)	32 (16.8)	19 (19.2)	0.605
Diabetes	24 (8.3)	13 (6.8)	11 (11.1)	0.207
Coronary heart disease	7 (2.4)	2 (1.0)	5 (5.1)	0.048
Cerebrovascular disease	2 (0.7)	1 (0.5)	1 (1.0)	1.000
Obesity	4 (1.4)	1 (0.5)	3 (3.0)	0.117
Allergic rhinitis	27 (9.3)	19 (9.9)	8 (8.1)	0.604
Asthma	9 (3.1)	1 (0.5)	8 (8.1)	0.001
COPD	0 (0)	0 (0)	0 (0)	NA
**Respiratory symptoms**				
Presence of symptom	121 (41.7)	71 (37.2)	50 (50.5)	0.029
Cough	64 (22.1)	36 (18.8)	28 (28.3)	0.066
Breathlessness	40 (13.8)	20 (10.5)	20 (20.2)	0.023
Sputum production	59 (20.3)	34 (17.8)	25 (25.3)	0.135
Wheezing	9 (3.1)	5 (2.6)	4 (4.0)	0.497
Sore throat	5 (1.7)	3 (1.6)	2 (2.0)	1.000
Chest tightness	7 (2.4)	4 (2.1)	3 (3.0)	0.694
Nasal obstruction	19 (6.6)	13 (6.8)	6 (6.1)	0.808
Runny nose	21 (7.2)	14 (7.3)	7 (7.1)	0.936

Data shown as n (%) or mean ± SD. COPD = chronic obstructive pulmonary disease; kg = kilogram; m = meter.

**Table 2 diseases-13-00372-t002:** Pulmonary function data and diagnosis of airway disease.

Variable	Data (n = 290)
**Spirometry data**	
FVC, L	2.94 ± 0.87
FVC, %predicted	92.2 ± 14.9
FEV_1_, L	2.41 ± 0.77
FEV_1_, % predicted	90.4 ± 15.9
FEV_1_ change after BD test, %	3.22 ± 4.61
FVC change after BD test, %	1.02 ± 5.94
FEV_1_/FVC, %	81.99 ± 8.40
FEV1/FVC, % predicted	97.94 ± 8.77
PEF, L/s	6.53 ± 1.97
PEF, % predicted	92.65 ± 19.40
FEF_25–75_, L/s	2.49 ± 1.18
FEF_25–75_, %predicted	85.75 ± 31.39
**Diagnosis of airway disease**	
Asthma	13 (4.5)
COPD	30 (10.3)

Data shown as n (%). BD = bronchodilator response; COPD = chronic obstructive pulmonary disease; FEV_1_ = forced expiratory volume in 1 s; FVC = forced vital capacity; FEF_25–75_ = forced expiratory flow at 25–75% of FVC; PEF = peak expiratory flow; L = liters; s = second.

**Table 3 diseases-13-00372-t003:** Abnormal pulmonary function and chest radiographic findings.

Characteristics	Data (n = 290)
Abnormal pulmonary function	99 (34.1)
Restrictive defect	48 (16.6)
Airway obstruction	26 (9.0)
Mixed obstructive and restrictive defect	8 (2.8)
Small-airway disease	17 (5.9)
Bronchodilator response	14 (4.8)
Abnormal chest radiographic findings	24 (8.3)
Lung fibrosis	3 (2.8)
Pleural fibrosis	6 (2.1)
Lung nodule	6 (2.1)
Mixed pattern	8 (2.8)
Silicotic pattern	1 (0.3)

Data shown as mean ± SD. Data shown as n (%); airway obstruction defined as FEV_1_/FVC ratio < lower limit of normal (LLN); restrictive defect defined as FEV_1_/FVC ratio > LLN and FVC < LLN; mixed obstructive and restrictive defect defined as FEV_1_/FVC ratio < LLN and FVC < LLN; small-airway disease defined as FEF_25–75_ < 65%; but normal FEV_1_, FVC and FEV1/FVC ratio; BDR defined as increase in FEV_1_ or FVC >10% of the predicted value after BDR test; BDR = bronchodilator response FEV_1_ = forced expiratory volume in one second; FVC = forced vital capacity; FEF_25–75_ = forced expiratory flow at 25–75% of FVC.

**Table 4 diseases-13-00372-t004:** Logistic regression analysis for factors associated with abnormal pulmonary functions.

Variables	Adjusted Odds Ratio (95%CI)	*p*-Value
Age for every 1-year increase	1.017 (1.001–1.033)	0.042
Smoking	1.002 (0.687–1.213)	0.657
Coronary heart disease	1.005 (0.564–1.126)	0.458
Pre-existing asthma	1.012 (0.987–1.135)	0.105
Presence of respiratory symptoms	0.798 (0.448–1.422)	0.244
Breathlessness	0.574 (0.265–1.243)	0.159

## Data Availability

Data is contained within the article.

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
