# Peer review of "Silicosis and Pulmonary Functions Among Residents Exposed to Dust in Saraburi Thailand"

_diseases, 2025, doi:10.3390/diseases13110372_

Round 1
Reviewer 1 Report
Comments and Suggestions for Authors
- Let's start with section 3.2, I quote: "In a previous study [29], the prevalence of silicosis among stone carving workers in 120 Nakhon Ratchasima Province, Thailand, was 36.1%. We hypothesized the same prevalence in our population. The sample size was calculated to estimate a proportion with 80% power, a type I error of 5%, and a precision margin of 5%, yielding a required sample size of 355." Please explain how these assumptions are actually impacting this study. What do you mean by Type I error? What does the letter (I) refer to?
- Still in methodology: Spirometry calibration procedures and inter-operator variability are not described, which limits reproducibility and reliability of pulmonary function results.
- The study should specify environmental dust concentration measurements to strengthen the exposure–outcome linkage.
- in line 209, why do you benchmark to the data in 1999 in Thailand, is there any more recent statistical on this to be used?
- The discussion lacks in depth analysis, it only reports on the observations. As this study aims to study silicosis, why the emphasis was turned mainly to Asthma? Kindply rpovide more in depth and balanced discussion.
- Please remove (we ) from all over the manuscript and use passive voice in alignment with academic language
- The conclusion has barely any content; it should be written to reflect the main findings (especially after providing a valuable in-depth discussion of the results, then move the limitation paragraph there, and then provide recommendations and future direction to be carried on.
Author Response
Comments 1: Let's start with section 2.3, I quote: "In a previous study [29], the prevalence of silicosis among stone carving workers in 120 Nakhon Ratchasima Province, Thailand, was 36.1%. We hypothesized the same prevalence in our population. The sample size was calculated to estimate a proportion with 80% power, a type I error of 5%, and a precision margin of 5%, yielding a required sample size of 355." Please explain how these assumptions are actually impacting this study. What do you mean by Type I error? What does the letter (I) refer to?
Response 1: Thank you for this valuable comment. We appreciate the opportunity to clarify the rationale and meaning of these statistical assumptions.
The sample size calculation was based on an assumed silicosis prevalence of 36.1% from a previous study conducted in a similar population. This assumption was used to ensure that our sample would be sufficiently large to estimate the true prevalence in our study population with acceptable precision and statistical confidence.
The Type I error (denoted as α) refers to the probability of rejecting the null hypothesis when it is actually true—that is, concluding that there is a significant difference or effect when in reality there is none. In this study, we set the Type I error rate at 5% (α = 0.05), which is a conventional threshold indicating a 5% risk of such a false-positive result. The letter “I” in Type I error simply represents the numeral one (1), indicating the first type of error in statistical hypothesis testing (Type I vs. Type II errors).
We have clarified this explanation in Section 2.3 (page 4 lines 157-161).
Comments 2: Still in methodology: Spirometry calibration procedures and inter-operator variability are not described, which limits reproducibility and reliability of pulmonary function results.
Response 2: Thank you very much for this important comment. We agree that describing the spirometry calibration and measures to minimize inter-operator variability is essential to ensure reproducibility and reliability. We have now added this information to the Methods section (pages 3 lines 114-118).
Comments 3: The study should specify environmental dust concentration measurements to strengthen the exposure–outcome linkage.
Response 3: Thank you for this insightful comment. We acknowledge that direct environmental dust concentration measurements would have strengthened the exposure–outcome relationship in our study. Unfortunately, measurements of ambient dust levels were not available during the study period.
To address this limitation and note the absence of quantitative exposure assessment data, we have added a statement to the Limitations section (page 10 lines 326-328).
Comments 4: in line 209, why do you benchmark to the data in 1999 in Thailand, is there any more recent statistical on this to be used?
Response 4: Thank you for this helpful comment. We agree that more recent national or regional data are preferable for comparison. We originally cited the 1999 Thai national survey because it remains the most comprehensive population-based study providing national prevalence and trend projections for COPD in Thailand. At the time of our analysis, no newer nationwide epidemiologic data were available beyond this report. However, we have also referenced more recent regional and institutional studies, including the report by Saiphoklang et al. (8.3%) in 2025 and a regional study in Chiang Mai (5–7%) in 2015, to provide updated context. We have clarified this rationale in the revised manuscript (page 10 lines 293-299).
Comments 5: The discussion lacks in depth analysis, it only reports on the observations. As this study aims to study silicosis, why the emphasis was turned mainly to Asthma? Kindly provide more in depth and balanced discussion.
Response 5: We agree that the discussion required more in-depth analysis and a clearer focus on silicosis. In the revised manuscript, we have expanded the discussion to provide a more balanced interpretation of the findings (pages 9-10 lines 264-282).
Comments 6: Please remove (we) from all over the manuscript and use passive voice in alignment with academic language.
Response 6: In accordance with the comment, all instances of first-person pronouns (e.g., “we,” “our”) have been removed throughout the manuscript. Sentences have been revised into the passive voice or restructured to ensure consistency with formal academic writing style.
Comments 7: The conclusion has barely any content; it should be written to reflect the main findings (especially after providing a valuable in-depth discussion of the results, then move the limitation paragraph there, and then provide recommendations and future direction to be carried on.
Response 7: We appreciate the reviewer’s constructive feedback. The conclusion section has been substantially revised to provide a concise summary of the main findings, reflecting the expanded discussion (page 11 lines 345-361).

Reviewer 2 Report
Comments and Suggestions for Authors
In diseases-3943891, Saiphoklang et al examine the epidemiology of silicosis and pulmonary functions among residents exposed to dust in Saraburi Thailand. The topic of this manuscript is interesting and fits well the scope of Disease. The reviewer feels it can be accepted after some amendments.
(1) The sample size is very small. Only 1 case of silicosis was observed. This is an obvious limitation of the study. The authors should address such issue.
(2) How did the authors assess the exposure to dust? This is a key issue.
(3) Table 1: Total (n=250); Table 2 and 3: 290, Results: Among 290 participants. Why the number is inconsistent?
(4) Amount of smoking, pack-years
2.7±8.6
8.4±13.6
11.5±14.5
0.329
Why total is even smaller than Normal lung function (n=179) /
Abnormal lung function (n=71)?
Author Response
Comments: In diseases-3943891, Saiphoklang et al examine the epidemiology of silicosis and pulmonary functions among residents exposed to dust in Saraburi Thailand. The topic of this manuscript is interesting and fits well the scope of Disease. The reviewer feels it can be accepted after some amendments.
Response: We would like to express my heartfelt gratitude to the reviewer for the wonderful reviews and comments.
Comments 1: The sample size is very small. Only 1 case of silicosis was observed. This is an obvious limitation of the study. The authors should address such issue.
Response 1: We have added this issue to the Discussion section (page 10 lines 325-326).
Comments 2: How did the authors assess the exposure to dust? This is a key issue.
Response 2: Thank you for this insightful comment. We acknowledge that direct measurements of environmental dust concentration would have strengthened the exposure–outcome relationship in the study. Unfortunately, ambient dust level measurements were not available during the study period. The only data available were from stone-grinding factories in Saraburi, Thailand, which included measurements of total dust and respirable crystalline silica, as cited in reference number 44 in the Discussion section (page 10 lines 304-307). We have also included this issue in the limitations of the study (page 10 lines 326-328).
Comments 3: Table 1: Total (n=250); Table 2 and 3: 290, Results: Among 290 participants. Why the number is inconsistent?
Response 3: We apologize for the mistake regarding the numbers. We have corrected the figures in Table 1 (total n = 290, normal n = 191, abnormal n = 99). Thank you for your thorough review.
Comments 4: Amount of smoking, pack-years
2.7±8.6
8.4±13.6
11.5±14.5
0.329
Why total is even smaller than Normal lung function (n=179) /
Abnormal lung function (n=71)?
Response 4: We apologize for the mistake regarding the numbers. We have reanalyzed and corrected the figure in Table 1 (total amount of smoking: 9.7 ± 14.0 pack-years). Thank you for your thorough review.

Reviewer 3 Report
Comments and Suggestions for Authors
This study conducted among residents in dust-exposed areas of Saraburi Province, Thailand, found a very low prevalence of silicosis (0.3%), while the prevalence rates of asthma and chronic obstructive pulmonary disease (COPD) were 4.5% and 10.3%, respectively. 34.1% of participants exhibited abnormal pulmonary function, with restrictive defects being the most common pattern (16.6%). Logistic regression analysis identified increasing age as a significant predictor of abnormal lung function. These results indicate that although silicosis is uncommon in this community population, dust exposure-related lung function impairment, particularly restrictive defects, is relatively prevalent. This highlights the necessity for targeted respiratory health surveillance and public health interventions in such environments.
My feedback is provided below.
- The abstract mentions that increasing age is a significant predictor of abnormal lung function, but it does not include several core findings. Could you please incorporate the specific types and prevalence rates of abnormal lung function, such as restrictive defect (16.6%) and airway obstruction (9.0%)? Additionally, the prevalence of abnormal chest radiograph findings (8.3%), including details like pulmonary fibrosis (2.8%) and pleural fibrosis (2.1%), should be included. The important observation that the prevalence of asthma and COPD (4.5%, 10.3%) was higher than previous community-based diagnosis rates, suggesting potential underdiagnosis, is also missing from the abstract.
- The text notes that data on silicosis and airway diseases in dust-exposed areas of Thailand are limited, but it does not explicitly state the study's key innovation. Could you clarify that this appears to be the first study in Thailand to investigate the prevalence of silicosis, asthma, and COPD, along with factors associated with lung function abnormalities, within a community-based population in Saraburi Province, rather than focusing on specific occupational groups such as stone carvers or ceramic workers?
- The research hypothesis is only mentioned in the Statistical Analysis section, where it states the assumption that the silicosis prevalence in this population is similar to the 36.1% found in previous studies of Thai stone carvers. For better logical flow, could this hypothesis be introduced earlier in the Introduction section?
- Regarding the review of previous studies, only two Thai occupational studies are cited. Are there data from other dust-exposed regions within Thailand, or from international community-level studies on dust exposure and lung function abnormalities, that could provide a more comprehensive context and better position this study within the wider research landscape?
- Several methodological details require further clarification. Could you specify the time frame used to define recent myocardial infarction and the criteria for determining inability to perform spirometry? Regarding dust exposure assessment, how was exposure intensity quantified, for example in terms of duration or concentration, and how was non-occupational environmental dust exposure evaluated? Furthermore, please describe the administration method for salbutamol during the bronchodilator response test and detail the quality control standards applied during spirometry.
- In the description of variable selection for multivariable logistic regression, the criteria of p<0.05 or significant effect on lung function is used. Could you please provide an objective definition for what constituted a "significant effect" in this context?
- For the regression analysis, were continuous variables such as age and pack-years of smoking treated as continuous or categorical? If categorical, what were the specific cut-off points used for stratification?
- The participant recruitment method is described as recruiting residents aged over 18 from Phra Phutthabat District. Could you please specify the sampling technique employed, for instance, whether it was random sampling or convenience sampling?
- Additional technical details would be beneficial. Could you provide information on the calibration frequency and standards for the Vyntus SPIRO spirometer, as well as the model and parameters for the chest radiography equipment? Please also describe the procedures for reviewing spirometry curves, such as whether a dedicated person performed this task, and any measures taken to ensure consistency in radiographic interpretation, for example inter-reader agreement statistics. Information on data entry quality control procedures would also be helpful.
- There appears to be a discrepancy in Table 1, where the header states n=250, while the main text reports a total sample size of 290. Could this please be clarified? Furthermore, for the identified silicosis case, a 37-year-old woman with 10 years in a cement factory, could you provide further clinical details such as her specific lung function indices and a more detailed description of her radiographic findings?
- I would suggest the inclusion of additional tables or figures to enhance the presentation of results. For example, a table showing the association between dust exposure history and types of lung function abnormalities, a bar chart illustrating the prevalence of lung function abnormalities across different age strata, and a histogram displaying the distribution of key lung function indices would be very informative.
- In the Discussion, the comparison of silicosis prevalence is currently limited to two occupational studies within Thailand. It would be valuable to also compare your findings on asthma and COPD prevalence, as well as lung function abnormality rates, with those from international community-based studies and global average prevalence rates.
- Regarding the study's limitations, could you please discuss potential unmeasured or residual confounding? For instance, were factors such as body mass index or exposure to biomass fuels considered, as these might influence lung function outcomes?
- Finally, I recommend including references to relevant studies that were not cited. These might encompass research from other dust-exposed communities in Thailand, international meta-analyses on community dust exposure and lung function abnormalities, and recent literature on advanced imaging techniques for silicosis diagnosis, which could help contextualise the limitation of using chest radiography alone.
Author Response
Comments: This study conducted among residents in dust-exposed areas of Saraburi Province, Thailand, found a very low prevalence of silicosis (0.3%), while the prevalence rates of asthma and chronic obstructive pulmonary disease (COPD) were 4.5% and 10.3%, respectively. 34.1% of participants exhibited abnormal pulmonary function, with restrictive defects being the most common pattern (16.6%). Logistic regression analysis identified increasing age as a significant predictor of abnormal lung function. These results indicate that although silicosis is uncommon in this community population, dust exposure-related lung function impairment, particularly restrictive defects, is relatively prevalent. This highlights the necessity for targeted respiratory health surveillance and public health interventions in such environments.
Response: We would like to express my heartfelt gratitude to the reviewer for the wonderful reviews and comments.
Comments 1: The abstract mentions that increasing age is a significant predictor of abnormal lung function, but it does not include several core findings. Could you please incorporate the specific types and prevalence rates of abnormal lung function, such as restrictive defect (16.6%) and airway obstruction (9.0%)? Additionally, the prevalence of abnormal chest radiograph findings (8.3%), including details like pulmonary fibrosis (2.8%) and pleural fibrosis (2.1%), should be included. The important observation that the prevalence of asthma and COPD (4.5%, 10.3%) was higher than previous community-based diagnosis rates, suggesting potential underdiagnosis, is also missing from the abstract.
Response 1: We have added the association between abnormal lung function patterns and age factors in the Results section (page 6 lines 205-207). Moreover, we have included the prevalence and underdiagnosis in the Abstract section (page 1 lines 33-34).
Comments 2: The text notes that data on silicosis and airway diseases in dust-exposed areas of Thailand are limited, but it does not explicitly state the study's key innovation. Could you clarify that this appears to be the first study in Thailand to investigate the prevalence of silicosis, asthma, and COPD, along with factors associated with lung function abnormalities, within a community-based population in Saraburi Province, rather than focusing on specific occupational groups such as stone carvers or ceramic workers?
Response 2: Thank you for your valuable comment. We agree and have revised the text to clearly highlight the novelty of our study. We have addressed these issues in the Introduction section (page 2 lines 80-86) and in the Discussion section (page 9 lines 237-240).
Comments 3: The research hypothesis is only mentioned in the Statistical Analysis section, where it states the assumption that the silicosis prevalence in this population is similar to the 36.1% found in previous studies of Thai stone carvers. For better logical flow, could this hypothesis be introduced earlier in the Introduction section?
Response 3: Thank you for your helpful suggestion. We agree that presenting the research hypothesis earlier improves the logical flow of the manuscript. Accordingly, we have added a statement in the Introduction section (page 2 lines 83-86).
Comments 4: Regarding the review of previous studies, only two Thai occupational studies are cited. Are there data from other dust-exposed regions within Thailand, or from international community-level studies on dust exposure and lung function abnormalities, that could provide a more comprehensive context and better position this study within the wider research landscape?
Response 4: Thank you for this helpful suggestion. We agree that expanding the literature review will better situate our study. We have expanded the Introduction to include additional relevant international community- and population-level studies on dust exposure and lung-function abnormalities (page 2 lines 71-79).
Comments 5: Several methodological details require further clarification. Could you specify the time frame used to define recent myocardial infarction and the criteria for determining inability to perform spirometry? Regarding dust exposure assessment, how was exposure intensity quantified, for example in terms of duration or concentration, and how was non-occupational environmental dust exposure evaluated? Furthermore, please describe the administration method for salbutamol during the bronchodilator response test and detail the quality control standards applied during spirometry.
Response 5: Thank you for your insightful comments. We have revised the Methods section to clarify the specific methodological details as suggested (pages 2-3 lines 95-96, 98-101, and 114-122).
Comments 6: In the description of variable selection for multivariable logistic regression, the criteria of p<0.05 or significant effect on lung function is used. Could you please provide an objective definition for what constituted a "significant effect" in this context?
Response 6: We appreciate the reviewer’s comment and the opportunity to clarify this point. In this context, significant effects on pulmonary function included age, smoking status, comorbidities, and respiratory symptoms (page 4 lines 165-168).
Comments 7: For the regression analysis, were continuous variables such as age and pack-years of smoking treated as continuous or categorical? If categorical, what were the specific cut-off points used for stratification?
Response 7: We appreciate the reviewer’s insightful question. In the regression analysis, age was treated as a continuous variable, while smoking status was included as a categorical variable (current/former smoker and never smoker).
Comments 8: The participant recruitment method is described as recruiting residents aged over 18 from Phra Phutthabat District. Could you please specify the sampling technique employed, for instance, whether it was random sampling or convenience sampling?
Response 8: We thank the reviewer for this valuable comment. We have addressed this issue in the Materials and Methods section (page 3 lines 98-101).
Comments 9: Additional technical details would be beneficial. Could you provide information on the calibration frequency and standards for the Vyntus SPIRO spirometer, as well as the model and parameters for the chest radiography equipment? Please also describe the procedures for reviewing spirometry curves, such as whether a dedicated person performed this task, and any measures taken to ensure consistency in radiographic interpretation, for example inter-reader agreement statistics. Information on data entry quality control procedures would also be helpful.
Response 9: We appreciate the reviewer’s detailed and constructive comments. Additional technical and quality control information has been included in the revised manuscript (page 3 lines 114-122 and page 4 lines 150-155).
Comments 10: There appears to be a discrepancy in Table 1, where the header states n=250, while the main text reports a total sample size of 290. Could this please be clarified? Furthermore, for the identified silicosis case, a 37-year-old woman with 10 years in a cement factory, could you provide further clinical details such as her specific lung function indices and a more detailed description of her radiographic findings?
Response 10: We thank the reviewer for carefully noting this discrepancy and for the request for additional clinical details.
We have corrected the figures in Table 1 (total n = 290, normal n = 191, abnormal n = 99).
Regarding the identified silicosis case, the details have been added to the Results section (pages 5-6 lines 196-201).
Comments 11: I would suggest the inclusion of additional tables or figures to enhance the presentation of results. For example, a table showing the association between dust exposure history and types of lung function abnormalities, a bar chart illustrating the prevalence of lung function abnormalities across different age strata, and a histogram displaying the distribution of key lung function indices would be very informative.
Response 11: We appreciate the reviewer’s thoughtful suggestions to enhance the presentation of the results. Unfortunately, detailed quantitative data on individual dust exposure history were not available in this community-based study; therefore, a table showing the association between dust exposure and lung function abnormalities could not be generated.
However, to improve data visualization and clarity, we have added additional figures as suggested. Specifically, a bar chart illustrating the prevalence of obstructive and restrictive lung function patterns across different age groups and a histogram displaying the distribution of FEV₁ and FVC (% predicted) have been included in the revised manuscript (page 5 lines 186-187, 190-191, and pages 7-8 Figures 2-4).
Comments 12: In the Discussion, the comparison of silicosis prevalence is currently limited to two occupational studies within Thailand. It would be valuable to also compare your findings on asthma and COPD prevalence, as well as lung function abnormality rates, with those from international community-based studies and global average prevalence rates.
Response 12: Thank you for this constructive suggestion. We have now expanded the Discussion to include comparisons of the prevalence of asthma, COPD, and lung function abnormalities with international community-based studies and global estimates (pages 9-10 lines 254-320).
Comments 13: Regarding the study's limitations, could you please discuss potential unmeasured or residual confounding? For instance, were factors such as body mass index or exposure to biomass fuels considered, as these might influence lung function outcomes?
Response 13: Thank you for your insightful comment. We agree that unmeasured or residual confounding may influence our findings. We have now added a discussion of potential confounders, including body mass index and exposure to biomass fuels, which were not fully assessed in our dataset but are known to impact respiratory health. This has been included in the Limitations section (page 10 lines 328-334).
Comments 14: Finally, I recommend including references to relevant studies that were not cited. These might encompass research from other dust-exposed communities in Thailand, international meta-analyses on community dust exposure and lung function abnormalities, and recent literature on advanced imaging techniques for silicosis diagnosis, which could help contextualise the limitation of using chest radiography alone.
Response 14: Thank you for this helpful suggestion. We have added citations and accompanying text comparing our findings with studies from other dust-exposed communities in Thailand, international meta-analyses evaluating the effects of dust exposure on lung function, and recent literature on advanced imaging modalities for silicosis. These additions strengthen the epidemiological context of our results and clarify the limitations of relying solely on chest radiography for silicosis screening (page 9 lines 269–274, and page 11 lines 338–341).

Round 2
Reviewer 1 Report
Comments and Suggestions for Authors
Now ok
Reviewer 3 Report
Comments and Suggestions for Authors
accept